# Surname affinity in Santiago, Chile: A network-based approach that uncovers urban segregation

**Naim Bro**[1], **Marcelo Mendoza**[1,2]*

**1** Millennium Institute of Foundational Research on Data, Santiago, Chile, **2** Department of Informatics, Universidad Técnica Federico Santa María, Santiago, Chile

* marcelo.mendoza@usm.cl

**Data Availability Statement:** All files are available from: Bro, Naim; Mendoza, Marcelo (2020): Surname affinity in Santiago, Chile: A network-based approach that uncovers urban segregation.

## Abstract

Based on a geocoded registry of more than four million residents of Santiago, Chile, we build two surname-based networks that reveal the city's population structure. The first network is formed from paternal and maternal surname pairs. The second network is formed from the isonymic distances between the city's neighborhoods. These networks uncover the city's main ethnic groups and their spatial distribution. We match the networks to a socioeconomic index, and find that surnames of high socioeconomic status tend to cluster, be more diverse, and occupy a well-defined quarter of the city. The results are suggestive of a high degree of urban segregation in Santiago.

## Introduction

Insofar surnames are associated with ancestry, they contain social information. Importantly, surnames can be a source of relational data. Novotný and Cheshire [1], for instance, construct a surname network from spatial proximity. In countries where individuals hold paternal and maternal surnames, last names can be used as direct relational data. For example, if Elena's paternal last name is González and her maternal last name is Muñoz, this is likely to mean that, at some point, a González and a Muñoz—her parents—had a relationship. Note that on some occasions this is not the case. For example, when people change their last names, or when the identity of the father is unknown.

This article uses surnames to produce a demographic synthesis of the population of Santiago, Chile. From records of more than four million people, we create two surname networks. The first one is based on paternal-maternal surname pairs and associates each last name to an approximate socioeconomic index. The second network represents the distances in the surname composition of urban locations, what the literature refers to as isonymic distances. For instance, if the residents of two urban areas possess similar last names, then the isonymic distance between these two locations will be small, and they will form a tie in the network. On the contrary, if two locations have very different surname compositions, they will possess a large isonymic distance and be disconnected.

figshare. Collection. https://doi.org/10.6084/m9.figshare.c.5230835.v1.

**Funding:** Naim Bro and Marcelo Mendoza acknowledge funding support from the Millennium Institute for Foundational Research on Data. Marcelo Mendoza was funded by the National Agency of Research and Development (ANID) grants Programa de Investigación Asociativa (PIA) AFB180002 and Fondo Nacional de Desarrollo Científico y Tecnológico (FONDECYT) 1200211.

**Competing interests:** The authors have declared that no competing interests exist.

Previous research has used last names to uncover the genetic [2–5], ethnic [6–8], and linguistic [9] composition of populations. For Chile, Barrai et al [10] use isonymic distances to reveal six macro-regions, organized along a North-South axis. Compared to previous research, our study innovates on two fronts. Whereas most previous research focuses on countries or world regions, we focus on a single city: Santiago. This distinction is important. At the country level, groups in geographically distant locations have limited possibilities for interacting. Therefore, their population structure is likely to reflect the availability of relationships in local areas rather than individuals' choices. Someone in a predominantly Hispanic region, for example, is likely to connect with a Hispanic person, not because she rejects connecting with non-Hispanics from a different region, but because this is what is available to her. In contrast, at the city level, the chances of between-group interaction are greater; therefore, the community structure that emerges is more likely to reflect who individuals choose to connect. Compared to previous research, the second innovation of our study is that it associates the surname data to a socioeconomic index. In addition to the paternal and maternal last names of Santiago residents, we know their approximate location in the income distribution. Thus, while previous research has focused on the ethnic and regional composition of populations, our data also allows us to assess the extent to which population structure is shaped by socioeconomic status.

We find that social status is an influential factor shaping population structure. In the paternal-maternal surname network, clusters of surnames associated to high-income groups emerge as clearly as ethnic clusters. Further, the analysis of isonymy reveals that surname composition is a significant marker to differentiate the high-income areas from the rest of the city.

## Materials and methods

### Data

The main source of data used in this study is the Chilean electoral registry of 2012, which contains the full name, the unique identifying number (R.U.T.) (RUT: *Registro Único Tributario), in Chilean administrative parlance.*, and the address of all individuals eligible to vote for political authorities in Chile (Persons over 18 years of age, including Chilean citizens and foreigners that have resided in Chile for more than five years). Only residents of Santiago were included in this analysis, totaling 4,652,933 individuals. The second source of data used in this study is the Territorial Well-being Index of 2012 [11], which indexes the mean income of every census administrative unit down to the block level, of which Santiago has 39901.

The data building phase involved geocoding every address in the electoral registry using the Google Maps API, which yields four types of definitions: approximate, geometric center, range interpolated, and rooftop. Only addresses geocoded with rooftop- and range interpolated-level precision were kept in the analysis, leaving 3,720,431 records. Then, each address was matched with a census block. Individuals' socioeconomic status was assigned based on the mean socioeconomic level of the blocks where they live. For example, if person A lives in block X, we define person A's socioeconomic status as the mean income of block X, as reported by the 2012 Territorial Well-being Index. Socioeconomic status was transformed into a 0-100 range using the formula $zi = \dfrac{xi - min(x)}{max(x) - min(x)}$, where $x = (x1, \ldots, xn)$ and $zi$ is the $i$th element of the normalized vector. The combined dataset includes the paternal and maternal surnames, the socioeconomic status, and the block identification for every person in the list. We share both networks with their respective data and attributes in a public repository. Please see https://doi.org/10.6084/m9.figshare.c.5230835.v1 to favor this study's reproducibility.

## Paternal-maternal surname network

We build an undirected network based on paternal-maternal surname affinities. The network is a graph $G(E, V)$ formed by a set of nodes $V$ and arcs $E$ that connect the nodes. Each node in $V$ represents a surname. We define the network using a weighted function over the arcs. An arc's weight is a positive integer representing the number of individuals who share a pair of surnames, irrespective of the paternal-maternal or maternal-paternal directionality. Arcs that do not seem to express affinity are dropped. For example, frequent Spanish surnames such as González or Muñoz may possess large frequencies. However, if their weight is less than expected given the surnames' large sizes, then it does not indicate affinity. In contrast, two rare surnames such Jadue and Manzur may have fewer connections but still more than expected given their small sizes. If the weight of an arc is larger than expected, then it is preserved.

Following Mateos *et al.* [12], we remove surname pairs if their weight denoted by $n_{ss}$ is less than a threshold $\frac{k \cdot n_{s1} \cdot n_{s2}}{N}$, where $k > 1$ is a parameter of the method, $n_{s1}$ is the number of occurrences of the surname $s_1$, $n_{s2}$ is the number of occurrences of the surname $s_2$, and $N$ is the number of individuals in the sample. Note that $\frac{k \cdot n_{s1} \cdot n_{s2}}{N}$ is $k$ times the expected number of co-occurrences of both surnames if the surnames are linked at random. Accordingly, $k$, is the security parameter of the method. A high value of $k$ generates a sparse network where ties are indicative of affinity. Mateo *et al.* [12] report that $k$ must be tuned to balance reliability and sparseness, allowing for a network representation that facilitates the detection of communities. Our empirical results are not sensitive to the estimation of the network's modularity for values of $k$ higher than 100. Accordingly, we used a threshold $k = 100$. Whereas the initial dataset contains 76969 unique surnames, after removing arcs with weights under $k = 100$, the network is left with 14115 nodes.

Next, we remove surnames that fall in the periphery of the network using a core analysis of the network. The $k$-core decomposition of the network [13] is the maximal subgraph that contains nodes of degree $k$ or more. The nodes that fall below a threshold $k = 3$ are removed. Thus, the network structure preserves the triangles that form the atomic units of relationships that characterize the networks of this type. After conducting this procedure, the network was left with 2106 nodes.

In the last step of the analysis, we use the Louvain algorithm [14] to detect the community structure of the network. Louvain is a modularity-based algorithm that maximizes the connectivity within detected clusters and minimizes the connectivity between groups. We ran the algorithm ten times and found similar numbers of communities (between 8 and 10) and modularity scores (between 0.566 and 0.581) across trials. We chose the most common partition, that of nine communities and a modularity score of 0.571, which is shown in Fig 1.

## Isonymy network

Isonymy was introduced by Lasker [15] to estimate the genetic relationship between populations. The method was then extended to analyze the geographical distribution of surnames in Britain [16]. The intuition is that areas are related if they share many surnames with similar relative frequencies. Isonymy $I_{i,j}$ is defined by $I_{i,j} = \sum_{k \in S} p_{k,i} \cdot p_{k,j}$, which is computed over the set $S$ of surnames that coexist in both areas, and $p_{k,i}$, $p_{k,j}$ are the relative frequencies of $k$ in the areas $i$ and $j$, respectively. Note that $I_{i,j}$ is a proximity function. If two areas have similar surname frequency distributions, and $S$ has many elements, $I_{i,j}$ takes large values. On the other hand, if both areas have few surnames in common, $I_{i,j}$ has values close to zero. If $S$ is empty, $I_{i,j}$ equals 0.

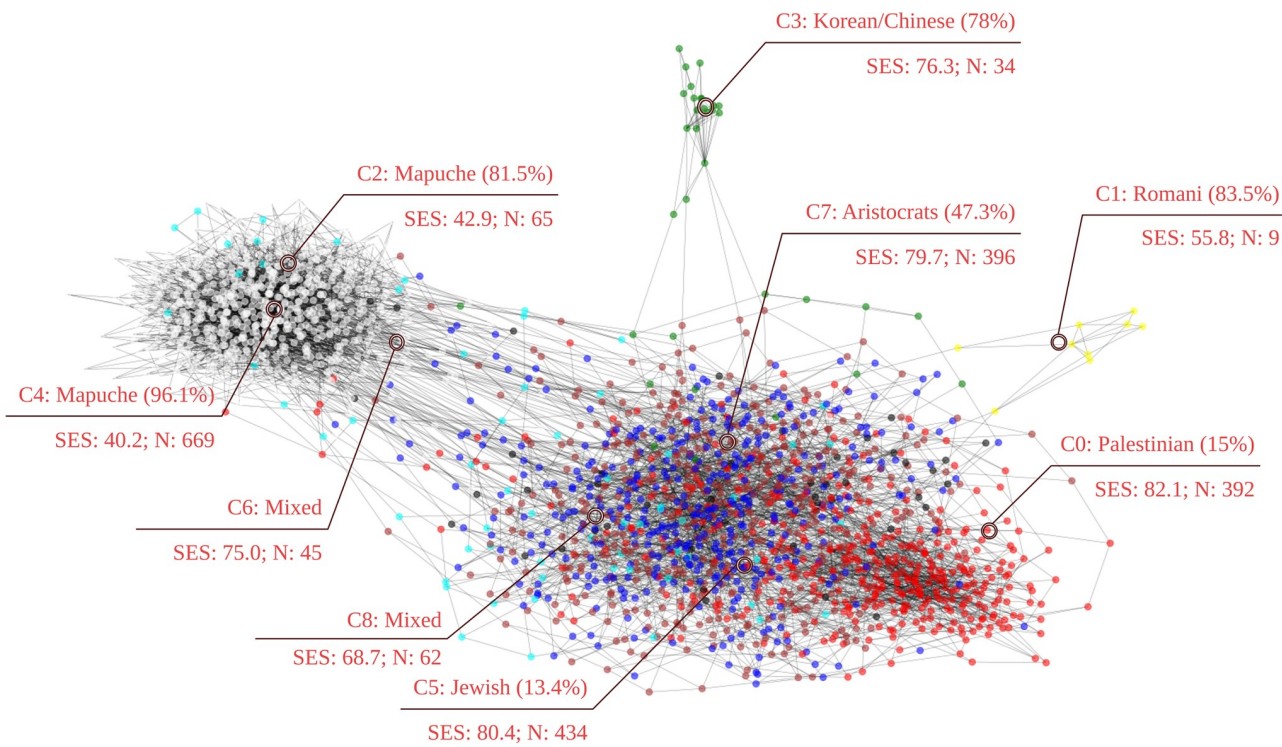

**Fig 1. Communities detected on the paternal-maternal surname network.** Some communities show a strong presence of surnames related to an ethnic group, while others are mixed. The two communities with the most significant presence of Mapuche are also those with the lowest SES (socioeconomic status) index. The communities with higher SES exhibit a strong presence of aristocratic, Jewish, and Palestinian surnames. Finally, some smaller communities show groups with little connection with the rest of society but a robust internal connectivity, as is the Romani and the Korean communities.

The relative frequency distribution of surnames provides information about the diversity of surnames in an area. Barrai *et al.* [17] proposed calculating $\alpha = \frac{1}{\sum p_k^2}$ to quantify the variety of surnames in a given area. This factor is also known as the effective surname number [18]. High $\alpha$ values indicate a diverse surname composition; low $\alpha$ values indicate homogeneity.

The areas included in the model are defined by segmenting the map of Santiago into a regular grid with 64 horizontal lines and 64 vertical lines. The resulting grid contains 4096 cells, each covering the same area. Of these cells, 621 match urban areas. The remaining cells were discarded for the analysis. The average alpha for the 621 areas is 267.6±126.5, which reflects more diversity than the average for the 54 provinces of Chile (209.2±8.9) as reported by Barrai *et al.* [10].

Isonymy is used to calculate a distance function between all pairs of urban locations in Santiago. The literature offers three isonymy-based ways for measuring a distance function between areas: Lasker's, Nei's, and Euclidian. Lasker's distance [19]$LD_{i,j} = -\log I_{i,j}$ is computed using the negative logarithm of the measure of isonymy. Nei's distance [20]$ND_{i,j} = -\log \frac{I_{i,j}}{\sqrt{I_i \cdot I_j}}$ measures the negative logarithm of isonymy in correspondence with the product of both areas' isonomies. Finally, Euclidean distance [21]$ED_{i,j} = \sqrt{1 - \sum_{k \in S} \sqrt{p_{k,i} \cdot p_{k,j}}}$ is computed from relative frequencies. We evaluate the

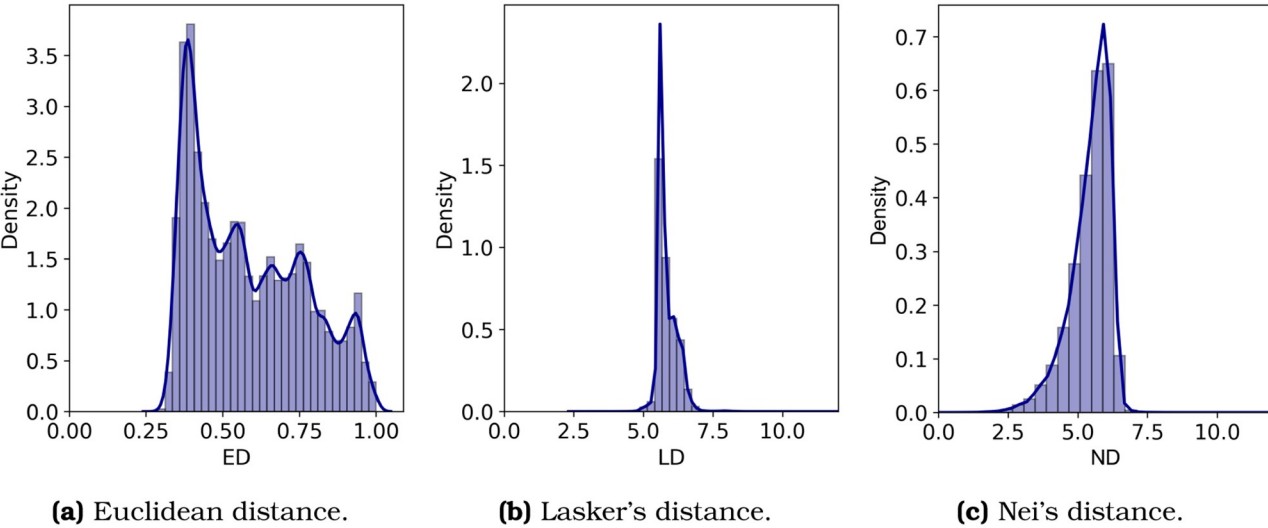

**Fig 2. Histograms of the three distance functions computed using the isonymy matrix.** The Euclidean distance (ED) shows better separability than the other two functions.

three functions to determine which differentiates the areas better. Fig 2 shows that the Euclidean distance (ED) produces a better separability between areas than the other two functions. Whereas ED shows a histogram ranging from 0.25 to 1.0, the other functions occupy a narrow segment of the distance domain. The mixture of normal distributions fitted to the data shows five components in ED, while LD and ND display only one mode. Since ED produces better separability between areas, we use ED to create the spatial isonymic network.

Next, we construct a network of surname relatedness between urban locations. The network $G(V, E)$ corresponds to an undirected graph, where $V$ is the set of urban areas, and $E$ is the set of edges with weights computed using the distance function. Since the resulting network is fully connected, and to reveal communities we need a sparse network, we remove non-meaningful ties. Following Shi *et al.* [22], we prune the network by iterating the Minimum Spanning Tree (MST) algorithm multiple times, a procedure that the authors refer to as Multiple Minimum Spanning Tree (MMST). The method quantifies the dissimilarity between successive MST networks using Schieber's D-value [23]. The iterations stop when no novel information is detected across consecutive MSTs. We run 250 MMST iterations and extract the D-value of each pair of successive iteration (Fig 3(a)). After the first iterations, the MMST adds redundant information, evidenced by the D-values' low variability. To avoid introducing redundant information, then, we build the network by aggregating the first 20 MSTs (Shi *et al.* [22] built theirs using the first 10 iterations).

The resulting network has 621 nodes, 12400 edges, a density = 0.064, an average degree = 39.9, and a diameter = 4. To extract the community structure of the network, we run the Louvain algorithm [14] ten times. All ten trials produced four communities and averaged a modularity of 0.323±0.07. We select the most common solution, that with fours communities and a modularity = 0.323. Fig 3(b) shows the effective surname number for the four communities, showing a clear structure of clusters.

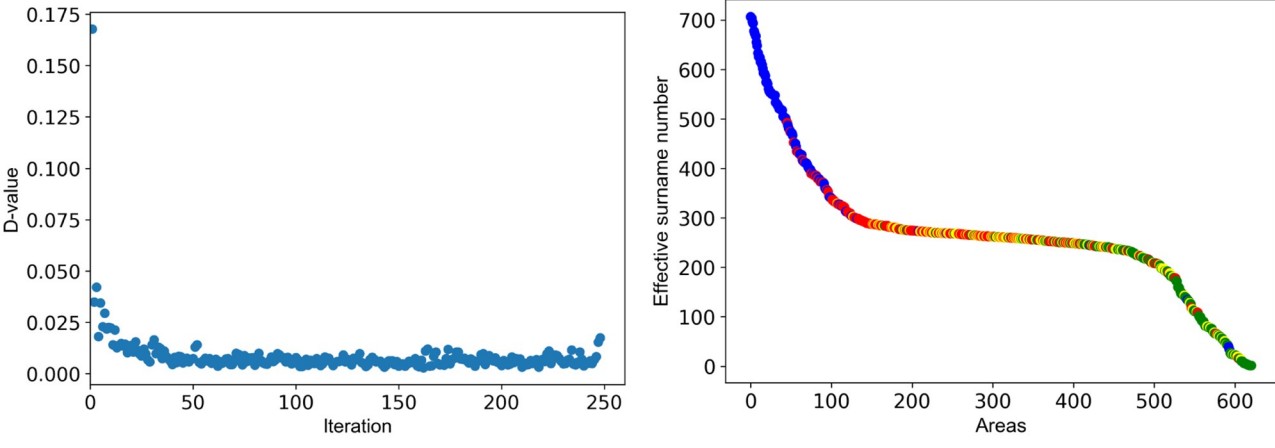

**(a)** D-values along 250 iterations of MSST.  **(b)** Effective surname number across areas.

**Fig 3. D-values and effective surname numbers.** a) 250 iterations of MSST were ran recording the D-values of every pair of consecutive MSTs; b) The effective surname number sorted in decreasing order show a segmentation of communities per $\alpha$.

## Results

### Paternal-maternal surname network

The community structure of the paternal-maternal surname network contains 9 clusters. In Fig 1, six of the nine communities can be linked to an ethnic minority. Clusters 2 and 4 are majority Mapuche, a native-Chilean people. They are formed by surnames such as Carilao, Lienlaf, Painen, and Curihuinca (see a sample of representative surnames per cluster in S1 Table). Our analysis identified two Mapuche communities that are located in different areas of the capital. Each of these communities produces intra-community interactions that generating crosses between families. We evidence a low interaction between the two communities, a factor attributable to the fact that they are located in different areas of the city. Cluster 0 has a substantial presence of Palestinian last names (Awad, Jadue, Hasbun), and Cluster 5 has Jewish last names (Ergas, Camhi, Cohen) mixed with some traditional upper class last names (Errázuriz, Aspillaga, Irarrázaval). The smaller clusters 1 and 3 are majority Korean (Lee, Kim, Park) and Romani (Nicolich, Savich, Aristich), respectively. The most representative surnames of cluster 7 are Edwards, Zañartu, Subercaseaux, etc, which contextual familiarity with Chilean surnames suggests represent the traditional upper class. Clusters 6 and 8 do not show the predominance of recognizable groups. The Palestinian (C0), Jewish (C5), and Aristocratic (C7) clusters present the highest mean socioeconomic status, and the two Mapuche clusters (C2 and C4) possess the lowest mean.

Another measure of status is each clusters' representation in politics. Fig 4 takes the list of all Chilean parliamentarians since 1830 [24], and for every period extracts the proportion that holds the surnames of our clusters. The aristocratic C7 is over-represented in the Chilean congress, especially in the nineteenth century. The partly-Jewish and the Palestinian clusters are also over-represented in politics. Clusters C1, C2, C3, C6, and C8 exhibit modest political representation. Notice that non-Hispanic last names were underrepresented in the nineteenth century, prior to the arrival of large numbers of migrants. The varied representation of the clusters in congress suggests that Chilean politicians are not randomly drawn from the population, but over-represent high-income communities.

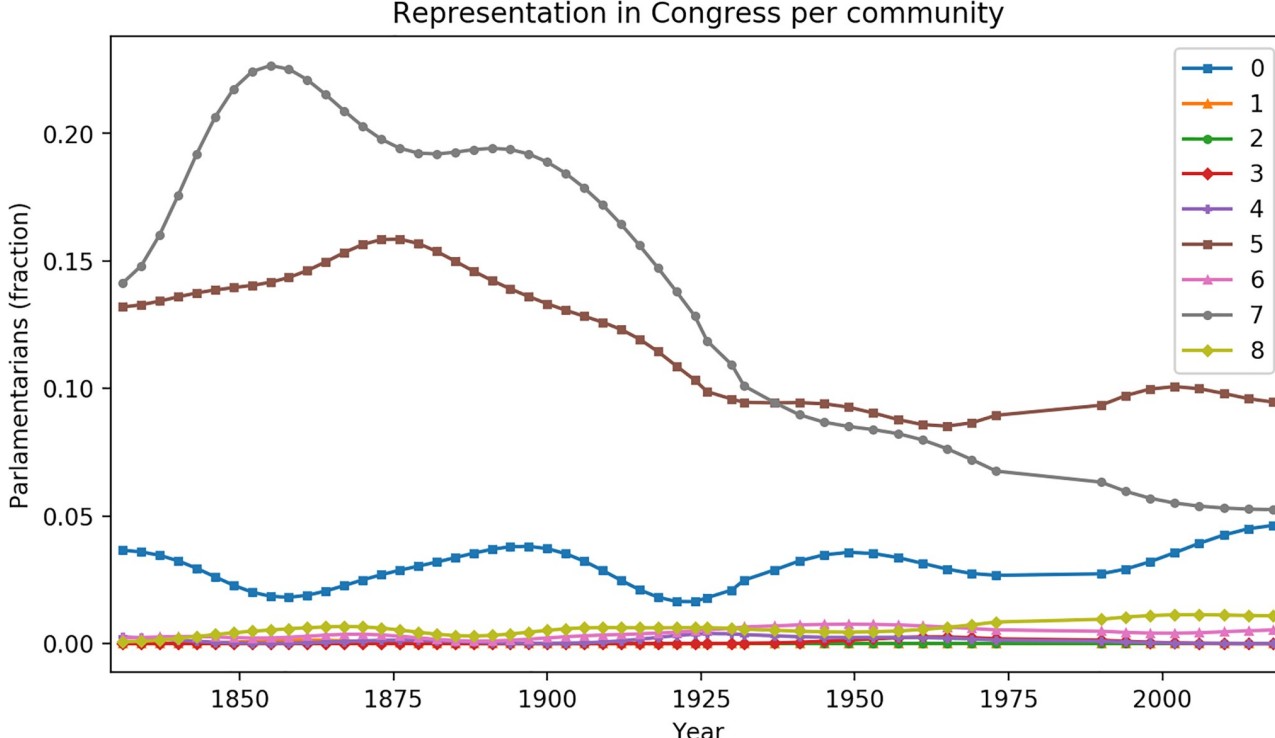

**Fig 4. Fraction of parlamentarians per community in the paternal-maternal surname affinity network.** Communities 5 and 7 had a salient representation in the 19th century, with a decline during the 20th century. Community 0 has had a presence since the congress's creation and shows a slight increase in its representation in the last decades. The other communities have meager representation.

Fig 5 shows the spatial distribution of people holding the representative surnames of each cluster. The "mixed" clusters (C6 and C8) are spread across the city, but the others are spatially concentrated. The Palestinian (C0), Jewish (C5), and Aristocratic (C7) clusters are located in the high-income North-East part of Santiago. The Korean cluster (C3) also has a North-East corner presence, but its primary locus is the city's center. The two Mapuche clusters (C2 and C4) occupy two main spots, in the North-West and the South. Finally, the Romani people tend to live in South Santiago.

### Isonymy network

The community structure of the isonymic network reveals four spatial clusters (see a sample of representative surnames per cluster in S2 Table). Fig 6(a) shows a cluster—depicted in blue—with a substantial isonymic distance to the rest of the network. The other three clusters—represented in yellow, red, and blue—are closer to each other. Fig 7 explores the structure of the network by visualizing the neighborhood of a sequence of high-degree nodes. We start with the highest degree-node of cluster blue, which connects exclusively with other same-cluster nodes, forming a network neighborhood that we call A (Fig 7(a)). Among the neighbors A, we chose the node with the highest degree and plot its neighbors B (Fig 7(b)). This second neighborhood is still mostly blue. We select the highest-degree node of neighborhood B and plot its neighborhood C (Fig 7(c)). While C is still mostly blue, more yellow dots appear. We chose the highest degree-node of neighborhood C and plot its neighborhood D (Fig 7(d)). Only in this

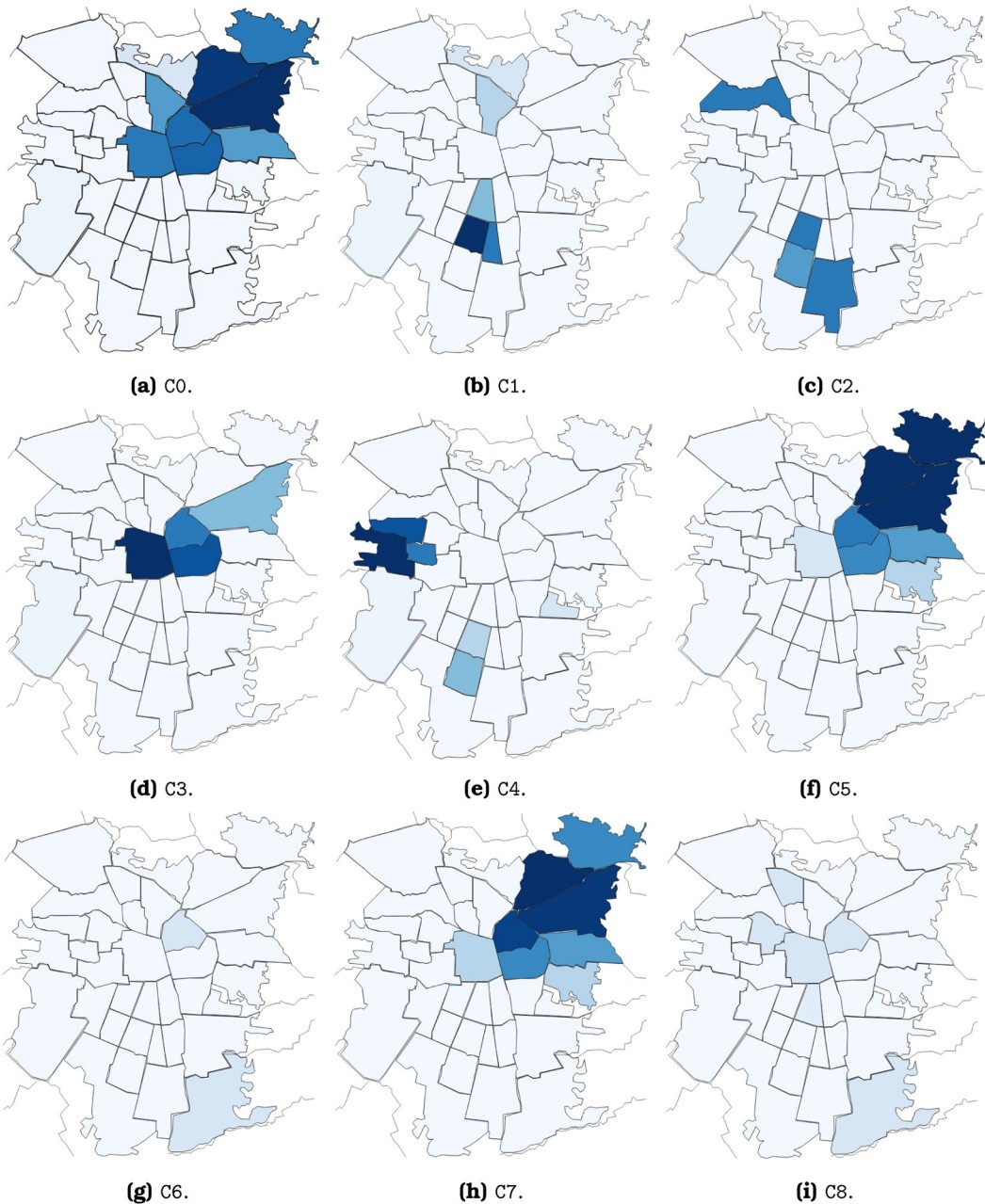

**(a)** C0.　　　　　　**(b)** C1.　　　　　　**(c)** C2.

**(d)** C3.　　　　　　**(e)** C4.　　　　　　**(f)** C5.

**(g)** C6.　　　　　　**(h)** C7.　　　　　　**(i)** C8.

**Fig 5. Spatial distribution of the communities detected on the paternal-maternal surname network.** Some communities are clearly associated to certain areas, while others are spread across the city. The map is shown segmented by communes, which correspond to the Metropolitan region's territorial political division. The colors indicate the presence of the community in each commune. Only communes with a presence of at least 10% of the indicated community have been colored. The more intense colors indicate a higher presence.

fourth step, we make a full jump from the blue cluster to the yellow cluster. In the fifth step (Fig 7(e)), we quickly find nodes belonging in multiple clusters, mostly yellow, red, and green. The last step (Fig 7(f)) walks into the periphery of the network, representing a majority of red nodes. The most remarkable result in this guided walk through the isonymy network's skeleton

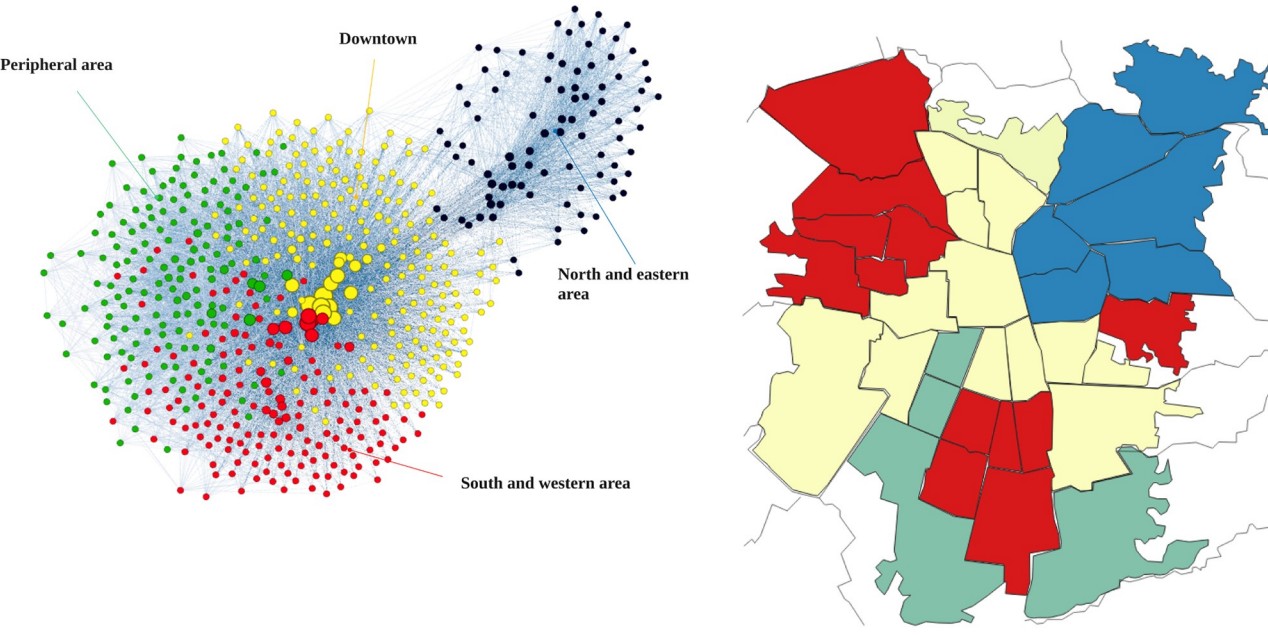

**(a)** The isonymy network.                                    **(b)** The spatial projection of the communities.

**Fig 6. The isonymy network and its spatial projection in Santiago de Chile.** a) The Isonymy network visualized using a Fruchterman-Reingold layout. The four communities are coloured showing a great separability among them; b) The spatial projection of the four communities in Santiago de Chile.

is that it takes four steps to make the jump from the blue cluster to the rest of the network. This reveals the isolation of the blue cluster relative to the others.

Further, the clusters are located in distinct parts of Santiago. In Fig 6(b), the blue cluster is concentrated in the high-income North-East corner of the map. While the other clusters are less patterned, the yellow cluster tends to be central, the green cluster tends to be peripheral, and the red cluster occupies spots in the North-West, the South, and to a lesser extent, the East part of Santiago.

The four clusters exhibit different values of surname diversity. In Fig 3(b), we see that the blue cluster presents a high effective surname number or $\alpha$. The yellow and red communities form a plateau on the $\alpha$ curve, and the green cluster exhibits low diversity. The average $\alpha$ per cluster are 509.1 (blue), 295.1 (red), 276.1 (yellow), and 262.2 (green). This result is consistent with Barrai *et al.* [10], who find that Chile's most diverse subnational unit is Vitacura, a commune located in our blue cluster.

Importantly, there is a positive association between the effective number of surnames $\alpha$ and socioeconomic status: the higher the income, the more diverse the surname composition, with a Pearson correlation coefficient of 0.54. In Fig 8, cluster blue exhibits a high socioeconomic status of 95.2 (a) and also a high $\alpha$ (b). Notice, however, that the association is not completely linear. In Fig 8(c), the effective surname number $\alpha$ is constant up to the 75th-percentile, and then it increases sharply. In other words, a 70th-percentile neighborhood is likely to resemble more a 10th-percentile area than a 90th-percentile one in terms of its surname composition.

The association between income and surname diversity is not unique to Chile. Collado *et al.* [25] find that rare surnames are over-represented in elite spaces in Spain. The reason, they argue, is that, in the past, socially ascendant individuals attempted to distinguish

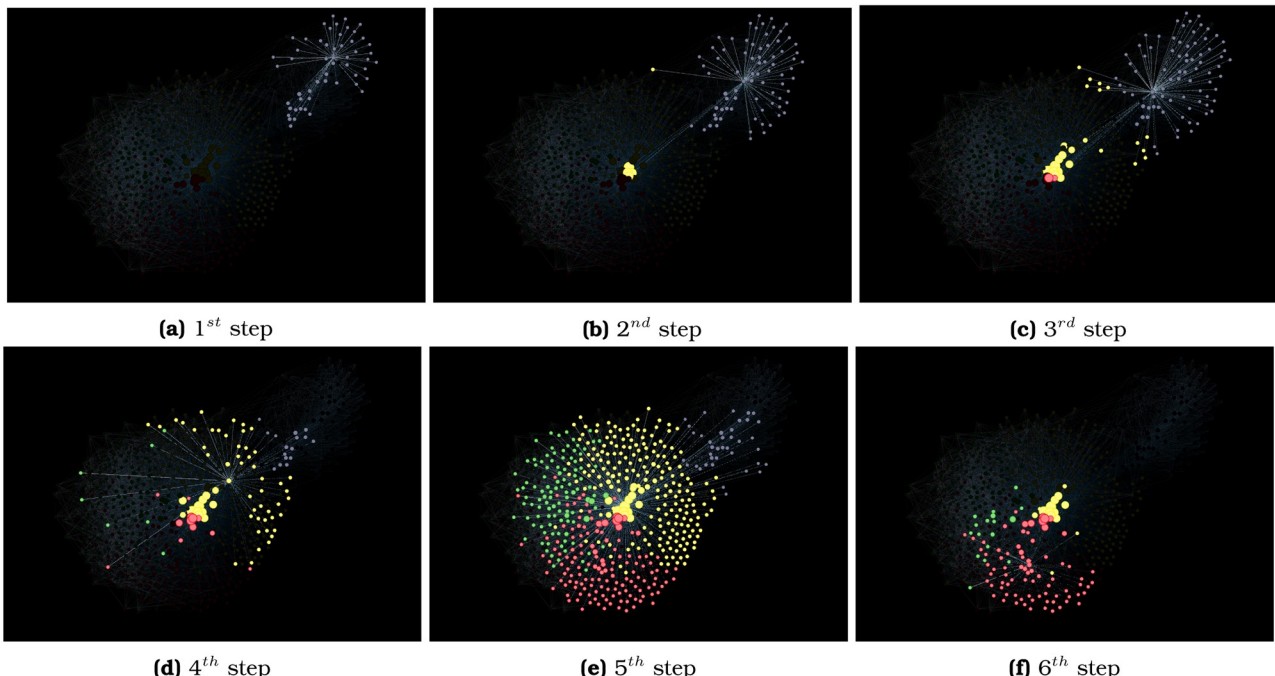

**Fig 7. Skeleton of the isonymy network.** Highly connected nodes uncover the skeleton of the network starting from the north eastern and ending in the periphery of the city.

themselves from the lower classes by double-barrelling their surnames. This resulted in their descendants, many of whom are still elites, holding rare names. Another reason why the elites may possess a diverse naming composition is that some immigrant groups are over-represented in the higher strata of society. We have seen, for example, that the Palestinian and

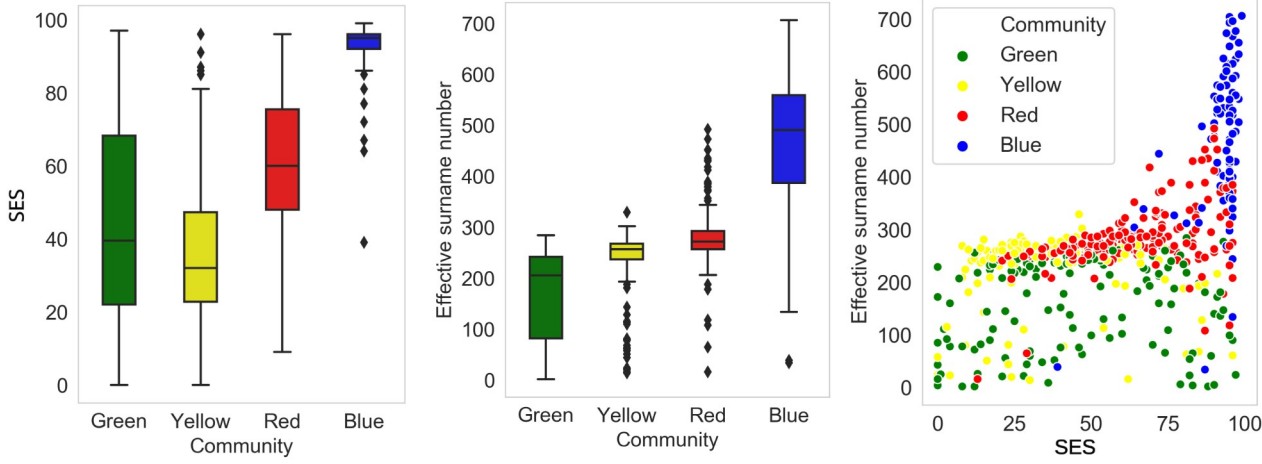

**(a)** SES distribution per community.   **(b)** $\alpha$ distribution per community.   **(c)** Scatter plot of SES and $\alpha$.

**Fig 8. SES and effective surname numbers within each community.** a) Distribution of SES per community; b) Distribution of the effective surname number per community; and c) Scatter plot between SES and the effective surname number.

Jewish clusters have a mean income comparable to that of the Aristocratic cluster. Their last names are rare in the overall population and diversify the surname pool of the elites.

The differences between the yellow, red, and green clusters are small but seem to respond to a pattern. If we compare Figs 5 and 6(b), we see that the red cluster corresponds to the areas were the Mapuche people live (see C2 and C4 for the predominantly Mapuche communities in Fig 5). The Mapuche people's presence in cluster red may explain its high diversity score compared to clusters yellow and green. In turn, the green cluster is the most homogeneous and the most distant from the city center. The fact that the periphery of Santiago has been settled more recently may account for the green cluster's relative homogeneity. The reason is that newly settled areas include only a subset of the surnames that exist in the communities of origin. This pattern is how Barrai *et al.* [10] account for why the North of Chile, which was settled by the Spanish colonialists first, is more diverse from a surname composition perspective than the country's South.

## Conclusion

This article maps the surnames of Santiago's residents into a relational and geographic space. The study takes advantage of a common practice in Spanish-speaking countries of citizens adopting both parents' last names. It is the first to associate surnames to an approximate socioeconomic index. This additional information allows uncovering the ethnic structure of the population as well as its social stratification. Another innovation of this study is exploring a city's surname structure instead of a country or macro-region. This distinction is relevant because the dwellers of a city are more likely to interact with each other than a country's inhabitants. Therefore, the population structure that emerges from their interactions is more reflective of individual *choices* than the *availability* of relationship options.

Like previous studies, we show that ethnic minorities tend to cluster. In Santiago, the larger minorities that group together are the native-Chilean Mapuche and the Palestinian, and to a lesser extent, the Jewish, Korean, and Romani communities. A less expected result is the addition of a large cluster of surnames associated with the traditional Chilean upper class. People holding these surnames have a high socioeconomic status and an oversized representation in politics. The emergence of this cluster suggests that members of the traditional upper-class behave like ethnic minorities in terms of their interaction patterns. This idea needs to be explored further in future, targeted research. If proved correct, then it will have implications for the study of social mobility.

Finally, the spatial distribution of surnames shows great variation between the high-income urban quarters, on the one hand, and low- and mid-income quarters, on the other. We detected four urban clusters, three of them fairly similar along socioeconomic status lines and surname diversity. The fourth cluster exhibits high socioeconomic status and a great diversity in its surname composition. Since surnames are shaped by ancestry and migration patterns, the association between socioeconomic status and the surname composition of populations requires attention in future research on urban stratification.

## Supporting information

**S1 Table. Top-10 surnames per community in the paternal-maternal surname affinity network.** Selected surnames are those with the highest degrees within each community.
(PDF)

**S2 Table. Top-10 surnames per community detected in the isonymy surname affinity network.** Prominent surnames are those frequent in an area but infrequent in the other areas.

Frequent and infrequent lists of surnames were computed using the top-500 lists of salient surnames in each area.
(PDF)

## Acknowledgments

Thanks to Josué Tapia and Andrés Cruz for helping in the data-building phase of this project. Other persons that helped are Daniel Alcatruz, Sebastián Huneeus, and Johans Peña.

## Author Contributions

**Conceptualization:** Naim Bro, Marcelo Mendoza.

**Data curation:** Naim Bro, Marcelo Mendoza.

**Formal analysis:** Marcelo Mendoza.

**Investigation:** Marcelo Mendoza.

**Writing – original draft:** Naim Bro, Marcelo Mendoza.

**Writing – review & editing:** Naim Bro.

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
