## [Decision Letter · Decision Letter 0]

28 Oct 2020

PONE-D-20-29865

Surname affinity in Santiago, Chile: A network-based approach that uncovers urban segregation

PLOS ONE

Dear Dr. Mendoza,

Thank you for submitting your manuscript to PLOS ONE. After careful consideration, we feel that it has merit but does not fully meet PLOS ONE’s publication criteria as it currently stands. Therefore, we invite you to submit a revised version of the manuscript that addresses the points raised during the review process.

We look forward to receiving your revised manuscript.

Kind regards,

Francesc Calafell

Academic Editor

PLOS ONE

Journal Requirements:

2.We note that you have stated that you will provide repository information for your data at acceptance. Should your manuscript be accepted for publication, we will hold it until you provide the relevant accession numbers or DOIs necessary to access your data. If you wish to make changes to your Data Availability statement, please describe these changes in your cover letter and we will update your Data Availability statement to reflect the information you provide.

4.We note that [Figure(s) 2 and 5] in your submission contain [map/satellite] images which may be copyrighted. All PLOS content is published under the Creative Commons Attribution License (CC BY 4.0), which means that the manuscript, images, and Supporting Information files will be freely available online, and any third party is permitted to access, download, copy, distribute, and use these materials in any way, even commercially, with proper attribution. For these reasons, we cannot publish previously copyrighted maps or satellite images created using proprietary data, such as Google software (Google Maps, Street View, and Earth). For more information, see our copyright guidelines: http://journals.plos.org/plosone/s/licenses-and-copyright.

1.    You may seek permission from the original copyright holder of Figure(s) [2 and 5] to publish the content specifically under the CC BY 4.0 license. 

Additional Editor Comments (if provided):

Bro and Mendoza do a great job of describing the surname relationships among the population of Santiago, the links with urban distribution and segregation, and with SES. Still, the paper falls short on the interpretation of its findings, and, while the conclusion hints to future analyses, some are needed in this paper. Just two examples:

* Why two Mapuche paternal-maternal surname clusters?

* Why the rich have more diverse surnames? It seems that the upper class contains different ethnic groups, and I suggest that the non-Chilean reader needs some context on how the richer Santiagoans have such diverse backgrounds. It could be contrasted with the also surname-diverse upper SES class in Spain, which forced diversity by seeking to differentiate their surnames from those of the commoners (Collado et al., investigaciones económicas. vol. XXXII (3), 2008, 259-287)

A few lesser issues:

* l. 60: How was SES transformed into a 0-100 range? Please provide the equation used

* l. 171-172: please add a table or graph with the mean and sd of the SES values per surname cluster

* l. 174-181 and Fig. 9: I was wondering whether 1830, which is the start point of the political analysis, precedes the arrival of some migrant groups, and thus may be biased against them

English:

* l. 84 sensitivity -> sensitive

* l. 92 this type's networks -> the networks of this type

* l. 97 similar number -> similar numbers

* l. 99 choose -> chose

Reviewers' comments:

Reviewer's Responses to Questions

**Comments to the Author**

1. Is the manuscript technically sound, and do the data support the conclusions?

Reviewer #1: Yes

2. Has the statistical analysis been performed appropriately and rigorously? 

Reviewer #1: Yes

3. Have the authors made all data underlying the findings in their manuscript fully available?

Reviewer #1: Yes

4. Is the manuscript presented in an intelligible fashion and written in standard English?

Reviewer #1: Yes

5. Review Comments to the Author

Reviewer #1: I think this is a well written and well presented paper. Data, methods and analysis are clearly set out and results are nicely presented. I only have one comment regarding Figure 2 - I assumed this was based on a Kernel Density Estimation, the results of which can be highly variable depending on the input parameters. I think these need to be consistent across the maps (I'm not sure if this is presently the case) and articulated. The color ramp is also problematic since it can exaggerate some parts of the distribution and is not color blind friendly. Take a look at colorbrewer.org for alternatives. You might also consider a more straightforward representation such as cell counts to show the same information.

6. PLOS authors have the option to publish the peer review history of their article (what does this mean?). If published, this will include your full peer review and any attached files.

Reviewer #1: No

---

## [Author Response · Author response to Decision Letter 0]

8 Dec 2020

We include a file with the response to reviewers. Please check the document, where we highlight how we address each comment of the editor and the reviewers.

---

## [Editor Report · Decision Letter 1]

9 Dec 2020

Surname affinity in Santiago, Chile: A network-based approach that uncovers urban segregation

PONE-D-20-29865R1

Dear Dr. Mendoza,

We’re pleased to inform you that your manuscript has been judged scientifically suitable for publication and will be formally accepted for publication once it meets all outstanding technical requirements.

Kind regards,

Francesc Calafell

Academic Editor

PLOS ONE
---

## [Editor Report · Acceptance letter]

15 Dec 2020

PONE-D-20-29865R1 

Surname affinity in Santiago, Chile: A network-based approach that uncovers urban segregation 

Dear Dr. Mendoza:

I'm pleased to inform you that your manuscript has been deemed suitable for publication in PLOS ONE. Congratulations! Your manuscript is now with our production department. 

Kind regards, 

on behalf of

Dr. Francesc Calafell 

Academic Editor

PLOS ONE